# Ultrafast Growth of Large Area Graphene on Si Wafer by a Single Pulse Current

**DOI:** 10.3390/molecules26164940

**Published:** 2021-08-15

**Authors:** Yifei Ge, Mingming Lu, Jiahao Wang, Jianxun Xu, Yuliang Zhao

**Affiliations:** 1Key Laboratory for Biomedical Effects of Nanomaterials and Nanosafety, CAS Center for Excellence in Nanoscience, National Center for Nanoscience and Technology, Chinese Academy of Sciences, Beijing 100190, China; geyf@nanoctr.cn (Y.G.); lumm2020@nanoctr.cn (M.L.); Wangjh2020@nanoctr.cn (J.W.); 2School of Future Technology, University of Chinese Academy of Sciences, Beijing 100049, China

**Keywords:** graphene, ultrafast growth, silicon

## Abstract

Graphene has many excellent optical, electrical and mechanical properties due to its unique two-dimensional structure. High-efficiency preparation of large area graphene film is the key to achieve its industrial applications. In this paper, an ultrafast quenching method was firstly carried out to flow a single pulse current through the surface of a Si wafer with a size of 10 mm × 10 mm for growing fully covered graphene film. The wafer surface was firstly coated with a 5-nm-thick carbon layer and then a 25-nm-thick nickel layer by magnetron sputtering. The optimum quenching conditions are a pulse current of 10 A and a pulse width of 2 s. The thus-prepared few-layered graphene film was proved to cover the substrate fully, showing a high conductivity. Our method is simple and highly efficient and does not need any high-power equipment. It is not limited by the size of the heating facility due to its self-heating feature, providing the potential to scale up the size of the substrates easily. Furthermore, this method can be applied to a variety of dielectric substrates, such as glass and quartz.

## 1. Introduction

Graphene, a two-dimensional honeycomb network crystal with monolayer atomic thickness, is favored by scientists all over the world for its excellent and unique optical, electrical, thermal, mechanical and other physical properties. Graphene can be used as an ideal material in fields such as multifunctional composites, organic optoelectronic materials, hydrogen storage materials, supercapacitors and microelectronic devices, electromagnetic attenuation materials and so on [1,2,3,4]. In 2004, single-layer graphene was first observed by Geim and Novoselov at Manchester University using the mechanical exfoliation method, which utilizes common cellophane tape to successively remove layers from a graphite flake. Despite its thermodynamic instability, graphene can exist stably in nature [5,6] and this discovery rapidly caused researchers to develop several preparation methods and technologies over the past two decades. Among various synthesis methods, chemical vapor deposition (CVD) has proven to be the most popular and effective method to obtain large area, high-quality and layer-number-controllable graphene [7,8,9,10]. There are also some other novel preparation methods for graphene [11], such as the newly reported electrochemical exfoliation method [12,13]. Doped graphene is used for stochastic biosensors and can be easily obtained by electrochemically exfoliating graphite rods immersed in a solution without the need for high temperature.

Currently, people tend to obtain large area graphene in an economical synthesis method to realize its promising industrial applications. The CVD method usually uses a single crystal Cu, Ni or alloy substrate, as well as polycrystalline metal substrate with specific crystal plane, to act as catalysts to promote the growth of graphene, then the grown graphene generally needs to be transferred to a target material. It is easy to cause defects such as wrinkles, breakages and contaminants in the transfer process, which greatly reduce the properties of the obtained graphene [14,15,16,17]. Moreover, it is difficult to obtain larger sized graphene by the CVD method due to the limitation of the furnace chamber. Another recently reported method, laser direct writing, has shown the potential to achieve large area or patterned graphene by the heating effect of laser. LinLi’s team at Manchester University used laser irradiation to directly produce a large area graphene film with a size of about 3 cm^2^ in 3–6 s on a glass substrate pre-coated with a 5–28-nm-thick Ni film covered with olive oil [18]. However, a high-power laser source of 16 kW was required to give a laser spot of 60 mm with uniformly distributed energy. It is costly to use such a device for industrial production and is very difficult to scale up.

In general, the current growth methods for large area graphene are mainly restricted by the film transfer process, the size of furnace chamber and the high-power heat source for industrial applications. In our previous work, an ultrafast quenching method was reported to make graphene functional Ni tips using a pulse current [19]. In this work, we further utilized a single pulse current to flow through the surface of a Si substrate coated with 5-nm-thick C and 25-nm-thick Ni for graphene growth. A large area and continuous few-layer graphene film was successfully generated on the surface of Si substrate within 2 s. As-grown graphene film proved to be highly conductive and covered the substrate fully.

## 2. Results and Discussion

The schematic diagram of the experimental set-up for ultrafast growth of large area graphene on Si wafer (300 nm of SiO_2_ on Si substrate) is shown in Figure 1a. The Joule heat produced by pulse current flew through the Ni coating on the surface of Si wafer was used to grow large area graphene at a high temperature instantly. As shown in Figure 1b, the Si wafer displayed dazzling white light during the heating process; Figure 1c is a wave of the optimal quenching conditions with a pulse current of 10 A and a pulse width of 2 s recorded by an oscilloscope. Figure 1d is the photograph of the quenched Si wafers with a size of 10 mm × 10 mm. Figure 1e is the SEM images of graphene film; some dendritic nickel islands and holes were formed and the wrinkle of graphene film is clearly visible in the illustration. The Raman optical image in Figure 1f shows that the surface of the quenched Si wafers was relatively uniform. Nine points evenly distributed on the surface of the sample were selected for Raman testing with a 514 nm laser. The corresponding Raman spectra are in Figure 1g. We found a D peak near 1358 cm^−1^, the G peak near 1581 cm^−1^ and the 2D peak near 2708 cm^−1^. Compared with the G and 2D peaks of single-layered graphene, the G peak of as-grown graphene film shifts to a lower wavenumber, while the 2D peak shifts to a higher wavenumber, which prove that few-layered graphene has been grown [20]. At the same time, the ratio of I_2D_/I_G_ is less than 1, also implicating that the graphene produced was few-layered. The above results prove that few-layered graphene is almost fully covered on the surface of the sample. The quality of as-grown graphene, as well as the morphology of the surface of quenched Si wafer, was similar to that of graphene synthesized by laser direct writing [18].

The sheet resistance of the quenched Si wafer was measured by four-probe resistance tester (Res Map 178, CDE, Cleburne, TX, USA). The calculation formula is as follows:(1)$$R_{□}=F\frac{\pi }{\ln2}\frac{U}{I}=4.5324\cdot F\frac{U}{I}$$

*F* is the correction coefficient. The Si wafer is square and the ratio of the sample lateral size to the probe spacing is 10; the correction coefficient is 0.927. The measured and corrected values of five sets of points were 50.3167 Ω·sq^−1^, 44.4257 Ω·sq^−1^, 50.0836 Ω·sq^−1^, 62.0398 Ω·sq^−1^, 62.751 Ω·sq^−1^, respectively. These resistance values are more than one order smaller than those of the graphene glasses synthesized by CVD (~1000 Ω·sq^−1^) [21] and four times smaller than the graphene synthesized by the solid-phase laser direct writing (~205 ± 19 Ω·sq^−1^) [22]. The average sheet resistance of the graphene film (53.9234 Ω·sq^−1^) showed a good electrical conductivity of the graphene-functionalized Si substrate, indicating the formation of large area and continuous graphene on Si. This agrees with the above Raman results.

Our method is simple and highly efficient. The Joule heat produced by a pulse current within 2 s does not need any high-power equipment used for growth graphene. More importantly, it is not limited by the size of the heating facility due to its self-heating feature, providing the potential to scale up the size of the substrates easily.

Raman spectra were obtained by a spectroscope integrated with a SEM to further analyze in situ the corresponding relationship between the surface composition and the microstructure of the film on the surface of quenched Si wafer. The optical microscope image (Figure 2a), the SEM image (Figure 2b) and the Raman mapping (Figure 2c) of the same area of the graphene film were recorded. A Raman map labelled in red, blue and green areas is used to represent the distribution of the same type of Raman spectra with the corresponding colours in Figure 2d. By comparing the micro-morphology in Figure 2a−c, the Raman spectrum with blue colour corresponds to the dendritic islands on the quenched Si wafer surface and the Raman spectra with red and green colour correspond to the holes on the quenched Si wafer surface. For the Raman spectra with green colour, the value of I_2D_/I_G_ is around 1, which proves bilayer graphene has been grown in most areas of the holes. The value of I_D_/I_G_ in the blue Raman spectrum is obviously bigger than that in the red and green Raman spectra, suggesting that few-layer graphene on the surface of dendritic islands has more defects than the graphene on the holes.

The element composition and distribution of the graphene film on the surface of the quenched Si wafer are analyzed by the SEM elemental mapping in Figure 3a,b. The graphene film was mainly composed of four kinds of elements, namely Ni, Si, O and C. The dendritic islands, composed of Ni and C elements, can be identified as several hundred nanometer nickel-rich particles covered with graphene film, taking the according Raman spectra (Figure 2c,d) into account together. The holes are mainly composed of Si, C and O elements, formed by thermal evaporation of Ni atom during heating process and the formation of few-layer wrinkled graphene film on the surface of Si substrates. Figure 3c,d show the C 1s and Ni 2p XPS spectra of the as-grown graphene film. For the C 1s region (Figure 3c), a peak at 285.3 eV is attributable to the surface contaminant carbon. Another peak at 284.7 eV can be assigned to the graphene. A peak at 285.7 eV is attributable to the Ni_3_C and two peaks at 288.9 eV and 286.6 eV can be assigned to hydroxy and carbonyl functional groups attached to the graphene film. For the Ni 2p region (Figure 3d), the Ni 2p_3/2_ peak is observed at 852.7 eV and 853 eV; the binding energies agreed with those of the metallic fcc-Ni and hcp-Ni. A peak at 853.5 eV is attributable to the Ni_3_C. A peak at 855.7 eV and a satellite peak at 861.3 eV are attributable to the NiO. We can conclude that the surface of the quenched wafer contains graphene, contaminant C, Ni, Ni_3_C and NiO; the peaks are consistent with results reported in the literature [23,24,25,26].

The corresponding AFM image of as-grown graphene film in Figure 3e shows that many dendritic islands and holes were generated on the surface of the coating film during the annealing process, which corresponds to the above SEM image in Figure 3a. By using a white light interferometer, the morphology of a 113 um × 113 um selected area is characterized in Figure 3f. The morphology was also very uniform, which indicates a large area of graphene film formed on the whole surface of the Si wafer.

Figure 4a shows Raman spectra of the graphene on Si wafer under different pulse currents. When the single pulse current increased from 6 A to 9 A with pulse width fixed at 2 s, the value of I_D_/I_G_ decreased gradually, indicating a reduction of defects in the graphene. However, single-layer graphene was only generated at some points of the quenched Si wafer surface at the current form 6 A to 8 A. At a current of 9 A, the value of I_2D_/I_G_ was around 1 and bilayer graphene with less defects were formed on nearly half of the quenched Si wafer surface. When the pulse current was increased to 10 A, the Si wafer was covered with few-layered graphene, indicated by the value of I_2D_/I_G_ smaller than 1. More importantly, the few-layered graphene film covered the full surface of Si wafer according to the results in Figure 1e,g. With increasing pulse width, the G peak shifted to a low wavenumber and the ratio of I_2D_/I_G_ was gradually decreased, which proved that the layer of as-grown graphene changed from single to multiple.

Figure 4b shows the Raman spectra of the graphene on Si wafer under the different pulse width. At a pulse current of 6 A, there is no obvious difference between the two types of Raman spectra with the increase of the pulse width from 2 s to 5 s. A small pulse current results in a low surface temperature of the Si wafer, which is not enough to reach the growth temperature of large area graphene, even if the pulse width is 5 s. At a pulse current of 10 A, few-layered graphene was generated at the pulse width of 2 s while no graphene was generated at the pulse width of 5 s due to damage of the Si wafer at high temperatures. To summarize, the optimal annealing condition of a pulse current flowing through the surface of the Si wafer is 10 A and the pulse width is 2 s.

The growth process of graphene film on the Si wafer by pulse current quenching could be revealed by comparing the microstructure and Raman spectra of the surface of the Si wafers before and after quenching under different pulse currents. Figure 5a shows the SEM image of the Si wafer successively coated with a 5-nm-thick C layer and a 25-nm-thick Ni layer before quenching; the surface is relatively flat. Figure 5b and c are the SEM and HR-SEM images of the Si wafer after quenching with a pulse current of 6 A and a pulse width of 2 s. Some light grey and black holes appear on the Ni layer in Figure 5b. Wrinkled graphene thin film was formed inside the holes in Figure 5c, which was also proved by the Raman mapping results in Figure 5d,e. The intensity of the Raman peaks of the green area was small, indicating that the graphene crystallinity was relatively weak on most of the surface of the Si wafer, where the morphology of the nickel layer was basically unchanged. Figure 5f is a HR-SEM image of the Si wafer after quenching with a pulse current of 10 A and a pulse width of 2 s. A large area of dendritic nickel-rich islands was formed due to the spheroidization of the nickel layer at higher temperatures. Few-layered graphene covered the whole surface of the Si wafer.

Based on the above results, the growth process and mechanism of graphene film are discussed. In the heating process of quenching, a pulse current flew through the Ni layer on the Si wafer. Nickel carbide is firstly generated by utilizing the produced Joule heat, then decomposing into nickel atoms and carbon atoms at a temperature above 400 °C; the nickel atoms evaporate and the carbon atoms are converted into graphene film at a temperature around 1100 °C [27]. Some light grey and black holes firstly appear, then they are expanded with the increasing temperature. Some dendritic nickel-rich islands with a width of several hundreds of nanometers finally formed on the Si wafer after massive evaporation of nickel atoms. The islands were covered with few-layered graphene with a few defects and the holes were mostly covered with bilayer graphene with more defects. We proved the similar mechanism for graphene growth on Ni tip in our previous paper [19], which is different from the traditional segregation mechanism.

## 3. Materials and Methods

### 3.1. Graphene Growth

In our experiment, 5-nm-thick carbon thin film (the purity of carbon target is 99.99%, Acs-4000-C4, ULVAC, Chigasaki, Japan) and 25-nm-thick nickel thin film (the purity of nickel target is 99.99%, Lab−18, Lesker, Jefferson Hills, PA, USA) were formed successively on the Si wafers (300-nm-thick SiO_2_ on Si wafer, 10 mm × 10 mm × 0.4 mm thickness) by magnetron sputtering method.

The coated Si wafers were quenched by the following ultrafast quenching methods developed by us. As shown in the schematic diagram of the experimental set-up in Figure 1a, as a heating unit, a pair of copper electrode sheets with a length of 30 mm, width of 1 mm and thickness of 0.05 mm were placed on both ends of surface of the coated Si wafer symmetrically. To ensure a good surface contact between the Si wafer and the copper electrode sheet, a ceramic block was placed on top of the electrode sheet, then the two ends of the ceramic block were fixed on the ceramic base with dovetail clips. The copper electrode sheets were connected to the external power supply through the vacuum electrode flange and the whole system was fixed in a vacuum chamber equipped with a molecular pump. After the vacuum reached 10^−5^ Pa, a single pulse (generated by a HMP4030 DC electrical station, Rhodes and Schwartz company, Munich, Germany) with a current of 10 A and a pulse width of 2 s was provided by an external power supply for ultrafast heating. As shown in Figure 1b, the coated Si wafer displayed dazzling white light instantly during the pulse current flew through the copper electrode sheets; the corresponding current pulse waveform was recorded by an oscilloscope in the inserted image in Figure 1b. The whole quenching process was very instantaneous and the waveform showed that the elapsed time of the pulse current raised to 10 A was about 10 ms and a drop to 0 A was also recorded at the same time. During the heating process, the applied pulse current flew through the conductive coating on the surface of the Si substrate, which produced a large amount of Joule heat to make the surface of the Si substrate rapidly reach a red-hot state and then the Si wafer naturally cooled down. It was estimated that the heating rate was closer to 1000 °C/s by calculation with the cooling rate.

### 3.2. Materials Characterization

Scanning electron microscopy (SEM, Hitachi S4800+EDS, Tokyo, Japan, 15 kV) studies were performed on as-grown graphene film on Si wafer surface to observe the surface morphologies and the corresponding element composition. Raman spectroscopy (Renishaw inVia plus, Wotton under Edge, UK, 514 nm excitation wavelength, 1 mm spatial resolution) was used to measure the characteristics of graphene. Surface topography images were obtained using an atomic force microscope (AFM, Bruker Multimode−8, Santa Barbara, CA, USA) and white light interferometer (BW-S501, Nikon, Tokyo, Japan). The graphene film was identified by using a RISE Raman system with a Sigma 300 SEM instrument (Carl Zeiss Microscopy Ltd., Jena, Germany). The laser wavelength was 532 nm. The Raman mapping of graphene film was carried out with a grid spacing of 200 nm and an accumulation time of 1 s at each spot. Ni 2p and C 1s XPS spectra were recorded using an X-ray photoelectron spectroscopy (XPS, PHI5300, Perkin-Elmer, Fremont, CA, USA). Al Kα radiation was used as an excitation source. The XPS analysis area was set to a diameter of 500 μm and the step size to 0.05 eV with a base pressure of 10−9 Pa during all measurements.

## 4. Conclusions

In this paper, we successfully grew a large area of graphene film on Si wafer using an ultrafast single pulse current quenching method. A single pulse current flew through the Ni layer on the surface of the Si wafer successively coated with a 5-nm-thick C layer and a 25-nm-thick Ni layer. During the process of quenching, nickel carbide was firstly generated at a lower temperature, then it decomposed into nickel atoms and carbon atoms with the temperature increasing (above 400 °C); the carbon atoms were converted into graphene at a higher temperature (around 1100 °C). The optimal quenching condition is a pulse current of 10 A and a pulse width of 2 s. The thus-prepared few-layered graphene film on the Si wafers proved to cover the substrate fully with a size of 10 mm × 10 mm. Moreover, it showed a high conductivity with an average sheet resistance of about 53.9 Ω·sq^−1^. Our method is simple and highly efficient and does not need any high-power equipment. The sample can be self-heated by the pulse current without the need for an additional heating facility to provide heat; this provides the potential to scale up the synthesized graphene film by adjusting the pulse current or pulse width applied to a larger size of different substrates. Furthermore, this transfer-free method can be applied to a variety of dielectric substrates, such as glass, quartz, sapphire and so on. The thus-produced graphene-functionalized various substrates can be used for sensors, electrical heating devices, biological scaffolds, and so on.

## Figures and Tables

**Figure 1 molecules-26-04940-f001:**
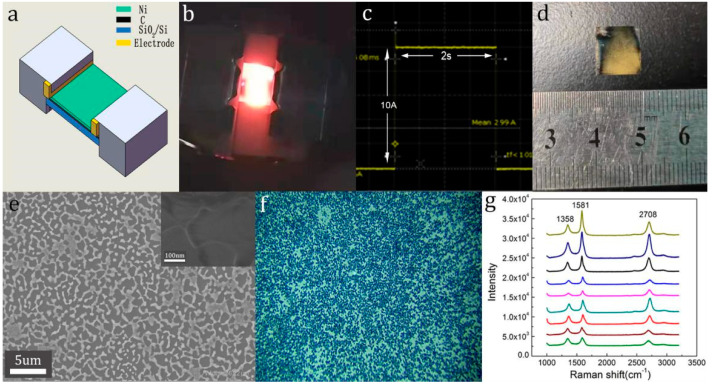
(**a**) Schematic diagram of experimental set-up for the growth of graphene film on the surface of Si wafer by ultrafast quenching method using pulse current; (**b**) photograph of Si wafer after a current flow through Cu electrode; (**c**) a single pulse of a current of 10 A with a pulse width of 2 s, recorded by an oscilloscope; (**d**) photograph of graphene thin film on the surface of Si wafer; (**e**) high-resolution SEM(HR-SEM) image of the graphene, the wrinkles of graphene film are clearly visible in the illustration; (**f**) Raman optical diagram of graphene thin film in Figure 1d; (**g**) Raman spectra recorded at different positions showed the fully covered graphene film.

**Figure 2 molecules-26-04940-f002:**
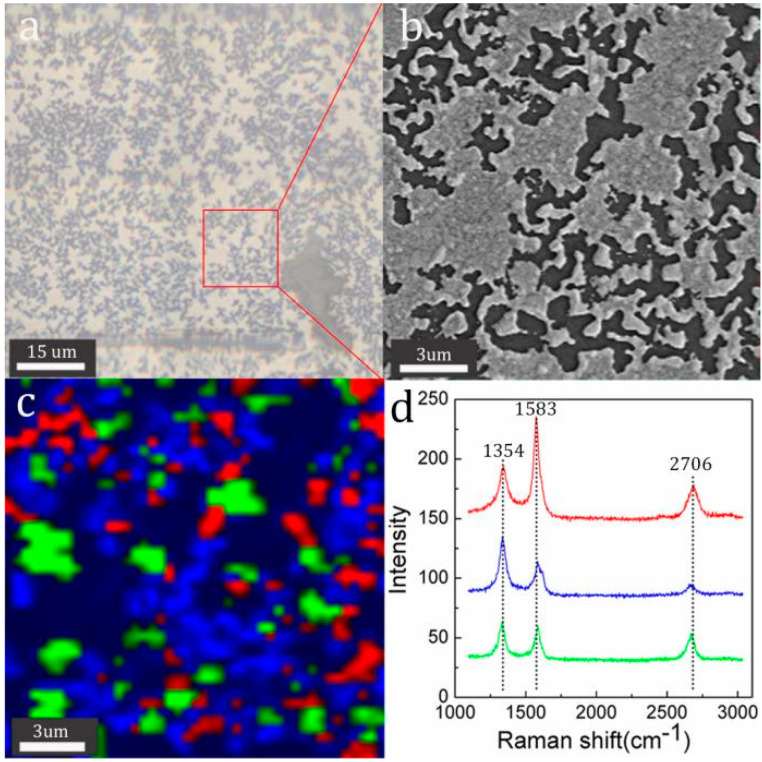
(**a**) Optical microscope image of the quenched Si wafer; the inserted image in Figure 2a is the corresponding selected area of SEM-Raman; (**b**,**c**) the SEM-Raman images ((**b**) is the corresponding SEM image and c is the corresponding Raman mapping); (**d**) three typical Raman spectra with different colours from the areas with corresponding colours in (**c**).

**Figure 3 molecules-26-04940-f003:**
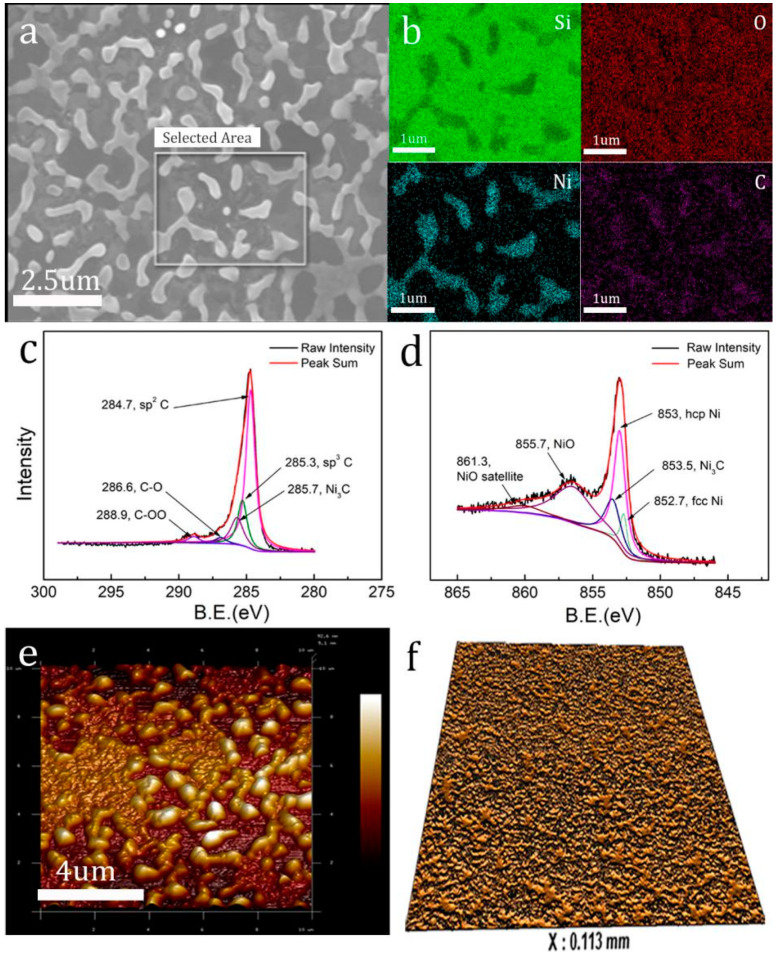
(**a**) SEM image of as-grown graphene film on Si wafer surface; (**b**) the corresponding SEM elemental mapping image, scale bar is 1 um. Green, red, cyan and purple dots indicate Si, O, Ni and C atoms, respectively; (**c**,**d**) XPS spectra of C1s and Ni2p for as-grown graphene film on Si wafer surface; (**e**) AFM image of surface morphology of the as-grown graphene film with a size of 10 um × 10 um; (**f**) image of surface morphology of the as-grown graphene film with a size of 113 um × 113 um by using white light interferometer.

**Figure 4 molecules-26-04940-f004:**
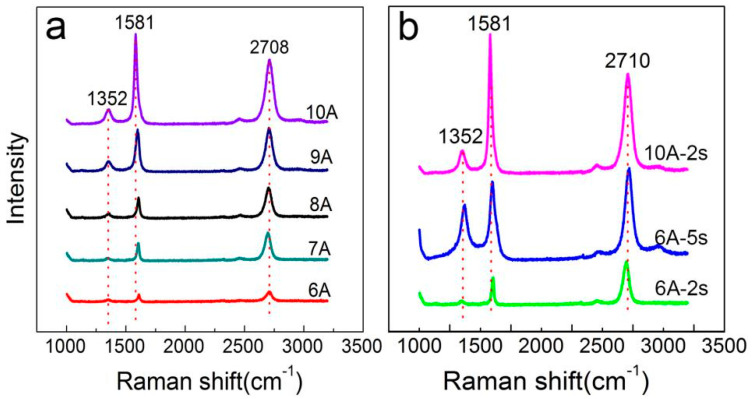
(**a**) Raman spectroscopy of the pulse current-dependent growth of graphene on Si wafer; (**b**) Raman spectroscopy of the pulse width-dependent growth of graphene on Si wafer.

**Figure 5 molecules-26-04940-f005:**
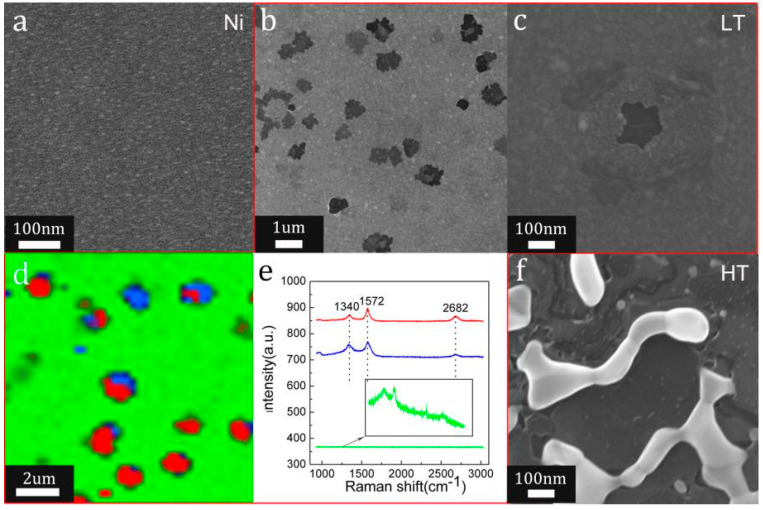
(**a**) SEM image of Si wafer coated with C and Ni layers before quenching; (**b**,**c**) the corresponding SEM and HR-SEM images after quenching with a pulse current of 6 A and a pulse width of 2 s; (**d**) Raman mapping of graphene; (**e**) Raman spectra of the red areas (red line), blue areas (blue line), green areas (green line) in (**d**); (**f**) HR-SEM image after quenching with a pulse current of 10 A and a pulse width of 2 s.

## Data Availability

The data presented in this study are available in the article.

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
