# Peer review of "Ultrafast Growth of Large Area Graphene on Si Wafer by a Single Pulse Current"

_molecules, 2021, doi:10.3390/molecules26164940_

Round 1

Reviewer 1 Report

The MS can be improved. The missing data are related to:

  1. comparison with other methods of synthesis of graphenes (please, check and cite the following: J. Electrochem. Soc. 168 037509, 2021; RSC Adv., 2021, 11, 23301–23309; Chemosensors 2021, 9, 146; Journal of The Electrochemical Society, 2020 167 037528);
  2. where you can apply the new synthesised graphene materials?

Author Response

Response to the referee 1:

Comment 1. Comparison with other methods of synthesis of graphenes (please, check and cite the following: J. Electrochem. Soc. 168 037509, 2021; RSC Adv., 2021, 11, 23301–23309; Chemosensors 2021, 9, 146; Journal of The Electrochemical Society, 2020 167 037528);

Response: Thanks for the comments. A description as bellow about other methods has been added in the first paragraph of Introduction part: “There are also some other novel preparation methods for graphene [11], such as the newly reported electrochemical exfoliation method [12,13], doped graphene was used for stochastic biosensors, which can be easily obtained by electrochemically exfoliating graphite rods immersed in a solution without the need for high temperature.”

Comment 2. where you can apply the new synthesized graphene materials?

Response: Thanks for the comments. This paper mainly introduces a new method for the growth of large area graphene films on different substrates. So-produced graphene functionalized various substrates are possible to be used for sensors, electrical heating devices, biological scaffold and so on. This point is also added to the Conclusion Part.

Reviewer 2 Report

The manuscript describes a new approach to fabricate large area graphene via single step pulse current method. The manuscript described a simple approach and characterizations such as SEM, AFM, and Raman of a few-layered graphene on Si substrate; however, I would recommend additional studies, for example, about a quality of graphene, thickness, and nature of defects for the publication of Molecules. The morphology shows that defect density is very high, therefore, I wonder if how the technique can prepare defect-free graphene in large area. In addition, it uses the joule heating through semi-conductive Si wafer. I also wonder if insulating substrates would work with this technique since it needs certain electrical resistance. From these considerations, I would recommend the rejection in the publication of Molecules.

Author Response

Response to the referee 2:

Comment 1. However, I would recommend additional studies, for example, about a quality of graphene, thickness, and nature of defects for the publication of Molecules. The morphology shows that defect density is very high, therefore, I wonder if how the technique can prepare defect-free graphene in large area.

Response: Thanks for the comments. It is the fact that the new method proposed in this paper currently cannot obtain graphene films without any defects. The XPS result has been added to the Fig.3, which indicates the hydroxy and carboxyl functional groups attached to the graphene. The following sentences have been added in the first paragraph on Page 4.“Figure 3c and 3d show the C 1s and Ni 2p XPS spectra of the as-grown graphene film. ……. We can conclude that, the surface of the quenched wafer contains graphene, contaminant C, Ni, Ni3C and NiO, the peaks are consistent with the results reported in the literature [23-26].”      

On the other hand, the Raman spectra have been compared with other references. The following sentences have been added in the first paragraph of the “Results and Discussion” part. “It is found that the D peak near 1358cm-1, the G peak near 1581cm-1 and the 2D peak near 2708 cm-1. Compared with the G and 2D peaks of single layed graphene, the G peak of as-grown graphene film shifts to a lower wavenumber, while the 2D peak shifts to a higher wavenumber, which prove that few-layed graphene has been grown [20]........The quality of as-grown graphene, as well as the morphology of the surface of quenched Si wafer, was similar to that of graphene synthesized by laser direct writing [18].”

Moreover, the synthesized graphene film still maintain a good electrical conductivity and it can cover the entire Si substrate within 2 s. This is proved by our electrical test results as shown in the second paragraph in Section 2 on Page 3.

Comment 2. In addition, it uses the joule heating through semi-conductive Si wafer. I also wonder if insulating substrates would work with this technique since it needs certain electrical resistance.

Response: We are sorry for any confusion on this point, it may be since we didn't clearly describe the heating principle. As explained in the first paragraph of the part of “results and discussion”on Page 2, the Ni and C layers were deposited on the Si wafer. Therefore, the Joule heat produced by pulse current flew through the Ni coating, instead of the Si or SiO2 wafer.

Reviewer 3 Report

In this paper, the authors present detailed discussions about the ultrafast growth of large area graphene on Si wafer by a single 2 pulse current.

The paper is well written, the English are poor.

The highlights are ok.

The paper has a lot of results, well explained but the authors don’t explain and don’t underline what is new in their work.

Some changes and clarifications are necessary:

  1. In the “Results and Discussion” figure 1b: please made a big picture of the oscilloscope recorder to see very clear.
  2. In the “Results and Discussion” figure 1f: please underline the graphene pick.
  3. In the “Results and discussion” part: please check your grammar (colour à color; spectrua à spectra/spectrum; grahene à graphene; obvious à obviously).
  4. Please replace Figure 2a (optical microscopy), with a focused image, and in Figure 2d please underline the graphene pick.
  5. Regarding your CVD technique, please compare your results with other results from the literature.
  6. In the sentence “The element composition and distribution of the graphene films on the surface of 118 quenched Si wafer are analyzed by the SEM-EDS results in Figure 3a-d.”, you have a mapping, not an EDS. Please be clear, if you talk about EDS please show an elemental spectrum.
  7. In the Conclusions part, please underline why your results are better than others obtained until now. What is new?

I recommend major revision.

Author Response

Response to the referee 3:

Comment 1. The paper is well written, the English are poor.

Response: Thanks for the comments. We have checked carefully for grammar and spelling mistakes and corrected them as shown in the manuscript.

Comment 2. The paper has a lot of results, well explained but the authors don’t explain and don’t underline what is new in their work.

Response: Thanks for the comments. We have added content to further highlight our innovations in second paragraph on page 3 as fellow: “This method is simple and highly efficient, the Joule heat produced by a pulse current within 2 s, which does not need any high-power equipment. More importantly, it is not limited by the size of the heating facility due to its self-heating feature, providing the potential to scale up the size of the substrates easily.”

Comments 3. Some changes and clarifications are necessary:

1.In the “Results and Discussion” figure 1b: please made a big picture of the oscilloscope recorder to see very clear.

2.In the “Results and Discussion” figure 1f: please underline the graphene pick.

3.In the “Results and discussion” part: please check your grammar (colour à color; spectrua à spectra/spectrum; grahene à graphene; obvious à obviously).

4.Please replace Figure 2a (optical microscopy), with a focused image, and in Figure 2d please underline the graphene pick.

5.Regarding your CVD technique, please compare your results with other results from the literature.

6.In the sentence “The element composition and distribution of the graphene films on the surface of 118 quenched Si wafer are analyzed by the SEM-EDS results in Figure 3a-d.”, you have a mapping, not an EDS. Please be clear, if you talk about EDS please show an elemental spectrum.

7.In the Conclusions part, please underline why your results are better than others obtained until now. What is new?

Response: 

  1. Thanks for the suggestion. It has been revised as shown in Figure 1.
  2. Thanks for the suggestion. We have compared the graphene picks with the results from the other literatures to support our conclusions.
  3. Thanks for the suggestion. The mistakes have been revised.
  4. Thanks for the suggestion. Figure 2a has been replaced, the graphene pick has been compared to other results.
  5. Thanks for the suggestion. The results have been compared with those from the literatures. The following sentences have been added in the first paragraph on Page 4.“Figure 3c and 3d show the C 1s and Ni 2p XPS spectra of the as-grown graphene film. ……. We can conclude that, the surface of the quenched wafer contains graphene, contaminant C, Ni, Ni3C and NiO, the peaks are consistent with the results reported in the literature [23-26].” On the other hand, the Raman spectra have been compared with other references. The following sentences have been added in the first paragraph of the “Results and Discussion” part. “It is found that the D peak near 1358cm-1, the G peak near 1581cm-1 and the 2D peak near 2708 cm-1. Compared with the G and 2D peaks of single layered graphene, the G peak of as-grown graphene film shifts to a lower wavenumber, while the 2D peak shifts to a higher wavenumber, which prove that few-layed graphene has been grown [20]........The quality of as-grown graphene, as well as the morphology of the surface of quenched Si wafer, was similar to that of graphene synthesized by laser direct writing [18].”
  6. Thanks for the suggestion. EDS has been revised as element mapping.
  7. Thanks for the suggestion. The advantages of our method have been further underlined in the Conclusions part.

Round 2

Reviewer 3 Report

The revised manuscript fulfills all the reviewer requirements.

I recommend to be accepted as it is.